# Impact of the 1918 Influenza Pandemic in Coastal Kenya

**DOI:** 10.3390/tropicalmed4020091

**Published:** 2019-06-08

**Authors:** Fred Andayi, Sandra S. Chaves, Marc-Alain Widdowson

**Affiliations:** 1Influenza Program, Centers for Disease Control and Prevention-Kenya, Nairobi 00621, Kenya; Fredandayi@gmail.com (F.A.); bev8@cdc.gov (S.S.C.); 2Influenza Division, National Center for Immunization and Respiratory Diseases, US Centers for Disease Control and Prevention, Atlanta, GA 30333, USA; 3Division of Global Health Protection, Centers for Disease Control and Prevention-Kenya, Nairobi 00621, Kenya; 4Division of Global Health Protection, Centers for Disease Control and Prevention, Atlanta, GA 30333, USA

**Keywords:** influenza pandemic, Spanish flu, 1918 pandemic, Kenya, Africa

## Abstract

The 1918 influenza pandemic was the most significant pandemic recorded in human history. Worldwide, an estimated half billion persons were infected and 20 to 100 million people died in three waves during 1918 to 1919. Yet the impact of this pandemic has been poorly documented in many countries especially those in Africa. We used colonial-era records to describe the impact of 1918 influenza pandemic in the Coast Province of Kenya. We gathered quantitative data on facility use and all-cause mortality from 1912 to 1925, and pandemic-specific data from active reporting from September 1918 to March 1919. We also extracted quotes from correspondence to complement the quantitative data and describe the societal impact of the pandemic. We found that crude mortality rates and healthcare utilization increased six- and three-fold, respectively, in 1918, and estimated a pandemic mortality rate of 25.3 deaths/1000 people/year. Impact to society and the health care system was dramatic as evidenced by correspondence. In conclusion, the 1918 pandemic profoundly affected Coastal Kenya. Preparation for the next pandemic requires continued improvement in surveillance, education about influenza vaccines, and efforts to prevent, detect and respond to novel influenza outbreaks.

## 1. Introduction 

The 1918 influenza pandemic or “Spanish flu” was likely the most significant global pandemic recorded in human history. An estimated 500 million people were infected and 20 to 100 million died in three successive waves between 1918 and 1919 [1]. In the United States most affected were healthy young adults, pregnant women and persons living in urban and peri-urban areas [2], though other risk factors such as poverty, underlying co-morbidities and malnutrition had a part of play and these factors varied in importance globally. The origins of the 1918 pandemic remain under discussion [3] but the first wave was initially recorded in the USA in the spring of 1918, spread within the USA, and was subsequently reported in Europe, and parts of Asia, causing high morbidity but relatively low mortality [4]. A second, more severe, wave came in the fall (September–December 1918) and caused exceptionally high rates of illness and deaths globally [5]. A less severe third wave appeared in early January 1919 and lingered on through March 1919 [6]. The countries that reported a first wave are thought to have experienced a reduced impact of the second wave compared to countries that did not report a first wave [7]. For example, India did not have a first wave and went on later to report a very high number of deaths of 10 to 20 million people in the second wave [6], more than all continental Europe combined [8]. Archived records from many countries, however, are often incomplete and vital statistic data are rarely robust. However, analysis of archived records from some countries, particularly from former British colonies, has provided insights on the pandemic, in many instances using a range of quantitative and qualitative sources [7,9,10,11,12,13,14,15,16,17]. 

In Africa, the pandemic has been described in South Africa, Ghana, Nigeria and Egypt [16]. In 1918–1919, after the First World War, all discharged soldiers were sent home, while others settled in newly acquired territories (colonies) overseas [3]. These included indigenous Kenyans of the supportive Carrier Corps, a military labor organization created by the British Administration in Kenya which recruited or conscripted over 400,000 African men for porterage and other support tasks during World War I. Some of these soldiers and support staff were infected and are thought to have introduced the pandemic influenza virus through seaports after which the virus spread inland along the railways, waterways and road networks to indigenous populations [15]. The seaports of Freetown, Sierra Leone in West Africa, Cape Town in South Africa, and Mombasa, Kenya in East Africa, have been identified as crucial pandemic entry points [15,16]. Ships from Aden, Yemen and India may additionally have introduced the influenza virus to British and French Somalia and Djibouti [15,16,18]. However, specific data on pandemic introduction and impact are very sparse from East Africa. 

Reportedly, a single ship from India which docked in Mombasa may have played a large role in introducing the virus to East Africa [19], which then spread to the entire Coast Province in Kenya and then to Zanzibar in Tanzania, and Entebbe in Uganda [16]. Lack of documentation from this era, along with current difficulties with assessing the burden of pandemic and seasonal influenza in East Africa, has hampered appreciation of the burden of influenza in this region [20,21,22]. In Kenya, the only report on the 1918 pandemic included aggregated mortality burden from routine reports by provinces, and estimated a pandemic mortality rate of 57.8 per 1000 people per year [13]. However, that report lacked baseline data, was based on passive reporting and did not describe the full impact of the pandemic including disruption to civil society. 

We turned to the Kenya National Archives (KNA) Library, Nairobi to investigate colonial-era records and correspondence, in order to describe better the magnitude and impact of 1918 influenza pandemic in the Coast Province comprising seven districts. We chose the Coast Province, because at that time, the region was the colony’s most critical administrative unit due to its important location on the coast, and maintained good administrative and health records. Moreover, the Province actively solicited data from districts in response to the arrival of the pandemic, providing a source of pandemic-specific data. Lastly, since vital statistics data from this period are incomplete and biased, we examined narratives of how the arrival of the pandemic in the Province affected people’s lives, providing additional insight into the health and societal impact of the influenza pandemic 

## 2. Methods

### 2.1. Study Area Background

Coast Province (previously referred to as Seyidie Province) was located immediately south of the Equator along 600 km of Indian Ocean coast, covering roughly an area of 150 by 450 km. It constituted seven administrative districts, namely: Mombasa (also referred to as Mombasa Island), located midway along the coastal stretch; Vanga (current day Kwale district) to the South; Taita Taveta to the West; Nyika (current day Kilifi); Malindi, North of Mombasa; and Tana River and Lamu districts in the North East. 

At the time of the 1918 pandemic influenza incursion, the capital of the British East African Protectorate (current day Kenya) had moved to Nairobi, but Europe, Arabia, India and the Americas sourced commercial goods from the Mombasa seaport. Therefore, those who settled in the Protectorate preferred the Coast Province, and this province had the largest non-native population in the Protectorate. The Mijikenda were the most predominant indigenous community in this region and were comprised of nine ethnicities: Giriama, Digo, Chonyi, Duruma, Jibana, Kambe, Kauma, Rabai and Ribe [23]. 

In the years prior to the pandemic, the Protectorate raised the hut tax to fund developmental plans (from 1 to 5 rupees in 1915, then to 8 rupees in 1920) [24]. In order to pay for this, the other taxes and the workforce needs of the colonial administration, the local populations chose salaried work (such as domestic help, farm laborers, or porters for colonial expeditions to the interior of the country) over subsistence farming [16,18,25,26]. This contributed to reduced food production, which caused malnutrition, social disruption and inter-tribe conflict [27]. Additionally, access to health care services was limited for most of the indigenous population. 

### 2.2. Data Collection and Analysis

#### 2.2.1. Health Information and Population Data, 1912–1925

In brief, we initially collected data on health facility use and all-cause mortality from published Protectorate Annual Reports and Coast Province Annual Reports, 1912 to 1925. However, when the data on facility use and mortality were missing, we searched and aggregated data from individual District Annual Reports for the same period. When these Annual Reports were missing data elements, we turned to facility reports of use and mortality complemented with reports from different district administrative departments such as social and cultural affairs or public health and sanitary reports and internal reports of mortality from weekly Barraza meetings (weekly community gatherings chaired by a local chief and council of elders to address issues affecting the local residents), outbreak reports, and security briefings. The contents varied by year and district, but all the data were part of the official disease and health facility use reporting to the district. When data were not available in a district-specific folder, we excluded that district from our summaries for that year (List S1). 

For denominator population, since no official national population census reports existed at that time, in order to calculate rates of health facility use and mortality, we obtained provincial denominator populations by year from Protectorate or Coast Province Annual Reports which had the same figures, and when unavailable (years 1917, 1918, 1919 and 1921), we used individual District Annual Reports and summed these to get a denominator for the Province.

For health facility use and mortality, we compiled data on the number of outpatient and inpatients using health facilities and all-cause mortality, from 1912 through 1925. We calculated overall health facility use and mortality rates by dividing the number of health facility visits or deaths in one district in one year by the population denominator for that district and year, and repeated this to arrive at province-wide rates. 

#### 2.2.2. Pandemic Influenza Data, 1918–1919

Following informal reports of the arrival of the 1918 influenza pandemic in the province, the Chief Medical Officer (CMO) and Provincial Commissioner (PC), jointly reached out to all district leadership, requesting them to monitor and regularly report on the then emerging outbreak. It remains unclear from the historical record when the first letter was dispatched to district offices, however, the earliest identified was to Malindi on 28 November 1918, although we assessed pandemic data received before this date which was sent in response to the unidentified earlier requests by Provincial authorities. These requests were for the number of influenza cases and deaths to inform the Province on the magnitude, impact and possible interventions strategies against the pandemic (Figure 1). During the following nine months, data were reported in a variety of summaries, in response to several reminders from the Province. Finally, on 10 June 1919, the Nairobi national government sent a letter to the Province asking for summary data on the pandemic by nationality from September to December 1918 which was forwarded to the district authorities (Figure 2). From these pandemic-specific reports from the districts, we extracted the number of reported pandemic cases and deaths with onset before 31 March 1919, after which no influenza-related deaths or cases were reported. 

The interactions between district and provincial level administrations on the pandemic generated a variety of files, which formed our study data sources. These included the minutes of local chiefs’ weekly Barraza meetings, health facility case and death summaries, case report forms, district officers’ correspondences letters and the routine district security/situational briefs. All these documents had been compiled by KNA into a Provincial special event folder named “Spanish Influenza KNA/PC/COAST/1/1/369”. Since this folder predominantly profiled mortality events, we sought additional data on all influenza-associated cases from Districts’ Annual Reports for the same period, in addition to files and reports from the Protectorate and Provincial administrations. The full list of documents used to gather specific data on pandemic influenza is listed in the Appendix A. 

Since we were fortunate to have a specific reporting system for pandemic mortality, we used the reported number of pandemic deaths as a numerator rather than to calculate excess mortality from the baseline routine all-mortality reports which were very low before the pandemic, likely due to underreporting, and increased in completeness for unknown reasons during and after the pandemic. We then calculated overall 1918 influenza pandemic mortality rates by dividing the number of pandemic deaths in a district to the districts’ denominator population, per 1000 individuals per year. For districts with reported pandemic cases (Mombasa, Taveta and Nyika), we also estimated the case fatality proportions (CFP), as a percentage proportion of reported pandemic deaths over the pandemic cases.

#### 2.2.3. Officials’ Quotes on Pandemic, 1918–1919

We examined the official documents, correspondence and extracted illustrative quotes on the overall burden, clinical symptoms, interventions, and social and economic disruption of the 1918 influenza pandemic, including comments on the general health situation and context of the affected communities. 

## 3. Results

### 3.1. Health Facility Use and All-Cause Mortality, 1912–1925

From 1912 through 1925, the entire Coast Province (seven districts) comprised a population of ranging from 170,000 to 243,841 (Appendix A). 

Prepandemic data of health facility use and mortality includes all the seven districts, except for years 1917, 1918, 1919 and 1921, which lacked data for both Lamu and Tana River districts; and years 1912 and 1917, which only had data from Mombasa district. Prior to the introduction of pandemic influenza in 1918, visits to healthcare facilities for outpatient or inpatient care varied between 8.6 and 32.6 visits/1000 persons per year, but increased five-fold to 146.8 visits/1000 person per year in 1918 (Figure 3). Similarly, trends of mortality from 1912 to 1925, showed relatively low recorded mortality before 1918 (0.5–4.0 deaths per 1000 people per year) followed by a sharp six-fold increase in 1918 (24 deaths per 1000 people per year) and continued high reported mortality (16.5 deaths per 1000 people per year) in the post-pandemic period through 1925 (Figure 4). 

### 3.2. Pandemic Influenza, 1918–1919

We focused on the five of the seven districts with data on pandemic cases or deaths and population denominators–Nyika, Vanga, Mombasa, Taita Taveta and Malindi. In the period 1918 to 1919, 181,199 residents lived in the five study districts, varying from 71,137 in Nyika to 19,761 in Vanga. The majority (92.5%) of the population were indigenous, followed by the Arabs and Asians (7.2%) and then Europeans (0.3%). Mombasa district contained most of the non-indigenous people (Asians, Arabs and Europeans) in the Province (Table 1). 

From September 1918 through December 1918, approximately 31,908 cases and 4593 deaths associated with 1918 pandemic influenza were reported from the five districts (Table 2). Nyika district reported the first cases in late September 1918, however the first fatal case was documented in Taita Taveta district on 1 November 1918. By 11 November 1918 all five districts had been affected. Mortality and morbidity varied considerably by district, although reports of the number of influenza cases were only available for three of the five districts and varied widely; from 1217 (33.0/1000) in Mombasa to 20,000 (281.1/1000) in Nyika, the largest district, and highest rates in Taita Taveta (380.0/1000). Similarly, mortality varied from 400 deaths in Vanga to 1700 in Nyika, and the rates varied from 17 in Mombasa to 35 in Taita Taveta and Malindi, the worst affected districts, for an overall mortality rate of 25.3 per 1000 people. For three of the five districts with available data on total reported influenza cases, we were able to calculate case fatality proportions. The total CFP was 10.6% (95%CI: 10.3–10.9), ranging from 8.5% in Nyika to 51.6% in Mombasa (Table 2, Figure 5). Some areas (e.g., some townships in Nyika) experienced little impact initially, but by November, all areas were reporting similar high severity. Duration of reported pandemic impact varied also. For instance, Nyika district experienced a prolonged epidemic course, until 20 March 1919 which may explain the overall larger reported burden. 

### 3.3. Burden and Clinical Syndrome of Pandemic Influenza

According to reports by District Officers and Medical Superintendents, pandemic cases presented with high fever, cough, headache, joint pains and were “refractory to treatment”, and these symptoms lasted up to eight days. An attending clinician in Taita, estimated that 30%–40% of the district’s residents were affected; and ~10% progressed to pneumonia and/or death. No data by age were available, but anecdotal evidence from Vanga district suggested that the elderly were relatively more affected. Of note, mortality estimates for the native population were challenging since this population was reluctant to provide information, yet reports suggest that the mortality may have been highest in these populations. Some example reports: 

“Similar accounts of the prevalence of influenza with so many cases complicated by pneumonia and consequent high mortality are reported throughout the province. Further medical assistance is not at present available.”CL Chevaliers, Mombasa District. Senior Medical Officer (Civil) 27 November 1918. [Reply letter to Coast Provincial Medical Officer on Spanish flu update: List S1. Spanish Influenza (1918–1919)-KNA/PC/COAST/1/1/369]

“Death occurred mostly among the old men and women, and judging from the number of elders of council reported to have died must have run into hundreds …and … Very few of the young and middle aged …Natives are most secretive about illness and death among their people”.RW Lambert Vanga Kakoneni Asst. District Commissioner (In absence of Ag.D.C) 20 January 1919. [Reply letter to a Senior Medical Officer of Health on Spanish flu update: List S1. Spanish Influenza (1918–1919)-KNA/PC/COAST/1/1/369]

“… at modest estimate between 30% and 40% of the Taita are down with the complaint. The death rate appears during the last few days to be approximately 50 per diem, and to be on the increase. …I anticipate at the present rate if the epidemic is of only 21 days duration the deaths will amount to probably over 1000, whatever action is taken”.Talbot Smith, Taita Taveta District Commissioner on 23 November 1918. [Reply letter to Coast provincial commission on Spanish flu update: List S1. Spanish Influenza (1918–1919)-KNA/PC/COAST/1/1/369]

“I consider the deaths have been augmented when either of the following two conditions have been present. (a) Overcrowding, as in Malindi, Mambrui and Roka. (b) Normally difficult conditions of life. I mean when food has been hard to come by or water far removed from villages. …north bank of the Sabaki River and in Chonyi”.Nyika District Commissioner 1 April 1919. [Reply letter to the senior medical officer on Spanish flu update: List S1. Spanish Influenza (1918–1919)-KNA/PC/COAST/1/1/369]

### 3.4. Public Health Interventions and Treatment

The colonial authorities developed guidelines for healthcare workers on precautionary measures; case management and convalescent care provision. According to this guideline, healthy persons were told to avoid contact with sick individuals and to take prophylactic remedies, such as throat gargling with potassium permanganate and other compounds, and oral quinine. Those already sick were advised to seek bed-rest, home nursing and proper nourishment, in addition to oral quinine three times a day. Health officials recommended aspirin for people with a headache and/or neurological signs. For the sick native population, the recommendations included home nursing, administration of paraffin oil (a teaspoon three times a day) and consumption of adequate starch and milk enriched meals. Lastly, in the convalescence phase, prolonged bed rest and slow return to work was advised, and for patients with depression, a tonic treatment was prescribed.

“… As regards to precautionary measures against the diseases, the avoidance of persons suffering from it, the frequent gargling of the throat (with solutions such as permanganate of Potash (half a grain to the pint), Chlorate of Potash (five grains to the ounces), Borax (ten grains to the ounces) etc.), the taking of small doses of quinine (two grains) morning and evening, keeping of ones dwelling well ventilated, and the living of a quiet life, keeping as much in the open air as possible are useful practices”.AD Milne Principal Medical Officer East Africa Protectorate on 21 November 1918. [Memo on precautionary measures against the Spanish flu to all Provincial and District medical officers in the East Africa Protectorate: List S1. Spanish Influenza (1918–1919)-KNA/PC/COAST/1/1/369]

“… The administration of one teaspoonful of paraffin oil three times a day has been spoken of as having good result when administered to Africans suffering from this disease. As regards the dieting of the sick, milk, uji (Porridge) of wimbi (millet meal), mtama (finger millet meal), or mahindi (maize meal), and such should be given at frequent intervals.”AD Milne Principal Medical Officer East Africa Protectorate on 21 November 1918. [Memo on precautionary measures against the Spanish flu to all Provincial and District medical officers in the East Africa Protectorate: List S1. Spanish Influenza (1918–1919)-KNA/PC/COAST/1/1/369]

“Possibly simple remedies which would give confidence without doing harm could be issued to chiefs for distribution such as quinine, salts or what not. My present object is to try and secure confidence and avoid possibilities of panic which I fear and which may be stopped if harmless faith giving remedies be supplied”.Nyika District Commissioner 14 April 1919. [Memo on precautionary measures against the Spanish flu to all Provincial and District medical officers in the East Africa Protectorate: List S1. Spanish Influenza (1918–1919)-KNA/PC/COAST/1/1/369]

### 3.5. Social and Economic Disruption

The colonial administration and local (native and non-native) residents experienced and reported different forms of social disruption. These ranged from paralyzed administrative operations, widespread food shortage, commercial losses, and overwhelmed healthcare sector (List S1). In four of the five District Administrative Offices, unprecedented absenteeism due to sickness during the pandemic period led to several disruptions of public service provision because of slowdowns, office closures and suspended operations. According to the colonial government correspondences, the acting Malindi District Commissioner reported that the pandemic influenza dramatically slowed government business operations. As one chief administrator noted, he was aware of one or more deaths occurring daily in the peak of the outbreak that affected all races. 

“Sir, I regret to report that the work of the district has been much retarded for the last ten days owing to influenza. The following Govt. servants have been off duty for the whole or part of the period…”SH Jadlau Malindi Ag. District Commissioner, 5 November 1918. [Letter to Provincial Commissioner reporting on the impact of Spanish flu: List S1. Spanish Influenza (1918–1919)-KNA/PC/COAST/1/1/369]

“…Influenza is getting no better, and we are having more deaths. On account of it, it is impossible to get porters, so we have had to write this morning to put off Dr. Shepherd’s Safari. He was to have brought me up the most pressingly needed drugs now they will be delayed.”Ada Drake, Sub Assistant Surgeon, Dabida Taita Taveta District, 25 November 1918. [Letter to District Commissioner requesting administrative support following a social disruption by Spanish flu incursion: List S1. Spanish Influenza (1918–1919)-KNA/PC/COAST/1/1/369]

For lack of options, the colonial authority contemplated soliciting natives’ carrier corps (porters) for the distribution of medical supplies, foods and information within the jurisdiction. A district commissioner at the time, Mr.Talbot Smith, is quoted in one of his correspondences to the Provincial Commissioner lamenting the magnitude of the pandemic problem. The social disruption and healthcare facility strain was reportedly similar across districts, irrespective of numbers of reported cases. Among the native population, the illness caused job losses; increased food insecurity and affected households’ ability to pay colonial taxes. Consequently, many natives suffered reduced “vitality” and low incomes. For those depending on subsistence farming, the total or partial crop failure occasioned by poor weather in 1918 and lack of seed supplies further exacerbated the poor health of the population. 

“Those locations where there has been a total or partial failure of crops, necessitating the inhabitants being on more or less half rations are suffering the most owing presumably to reduced vitality. …These natives are now suffering from reduced vitality and possibly cannot at least for the moment pay for assistance, as well as hut tax.”Mr Talbot Smith, Taita Taveta Voi, District Commissioner 25 November 1918. [Letter to Provincial Commissioner reporting on the impact of Spanish flu: List S1. Spanish Influenza (1918–1919)-KNA/PC/COAST/1/1/369]

Economic disruption was also reported in large commercial farms (e.g., Kedal and Hauber Estates-Farms), which suffered massive losses due to unprecedented labor shortage. In the healthcare sector, the situation was grave; understaffed facilities were overwhelmed with the influx of patients, low reserves of medical supplies and little colonial administration support. For instance, in Taita Taveta, at the peak of the outbreak, the local healthcare facilities were overwhelmed by the patient influx, with up to 300 cases seen daily. The enormity of the problem caused panic to an extent that the authorities allowed for the use of placebo therapeutics to pacify residents anger. 

“The Kedal Fibre Estate, BEA Corporation Estate (Farms) and the Haubner Estate (Farms) have virtually had to close down. The sub assistant surgeon seems to have run out of the necessary medicine owing to the abnormal drain and I trust you will find your way to supply whatever is necessary”.Mr Talbot Smith, Taita Taveta Voi, District Commissioner 23 November 1918. [Letter to Provincial Commissioner reporting the impact of Spanish flu: List S1. Spanish Influenza (1918–1919)-KNA/PC/COAST/1/1/369]

“Miss Drake to my knowledge has been treating 300 and more patients and would treat nearly 3000 if remedies had permitted”.Mr Talbot Smith, Taita Taveta Voi, District Commissioner 27 November 1918. [Letter to Provincial Commissioner reporting the impact of Spanish flu: List S1. Spanish Influenza (1918–1919)-KNA/PC/COAST/1/1/369]

## 4. Discussion

We used historical records and colonial officials’ correspondences in pre-independence Kenya, preserved at Kenya National Archives Library, to describe the magnitude and impact of the 1918 Influenza pandemic in the Coastal Kenya Region. In 1918, this region was Kenya’s most critical administrative unit, with its highest immigrant population proportion (European, Asian and Arab) and a key commercial hub. This study adds to the sparse data from Africa, and provides substantial insights from the most devastating infectious disease event in recorded history, suggesting that Coastal East Africa was as badly or worse affected by influenza as other parts of the world.

We noted, that in 1918, the crude death rates and healthcare utilization drastically increased, six- and three-fold, respectively and stayed relatively high until at least 1925. The sharp increase in health care utilization was certainly due to the pandemic and is corroborated by the anecdotal reporting of overwhelmed health systems. The very large majority of these cases would have been in the native population, though we had no data on race. The higher rates of mortality and facility visits after 1918 compared to before 1918 were likely due to improved reporting health facility expansion rather than prolonged pandemic transmission. Equally, it is plausible that several documented outbreaks such as the plague (1920) and smallpox (1925) [28,29], also contributed to high reported mortality and morbidity in those late years studied. We estimate pandemic mortality from September 1918 to March 1919 to be approximately 25 deaths/1000 population and morbidity at 176/1000 population or an attack rate of 17.6%.

In addition to health care and mortality burden, we document the serious social impact of the pandemic on society, both in the running of the colonial administration and the health of the population coping with profound societal disruptions linked to colonization [18,30,31], failure of agriculture and food shortages. These factors may have led to the native population even being more susceptible to the virus than others [26]. Some differences in the impact of the disease were also noted between areas. To some extent, this was attributed at the time to differences in living standards and exposure to the outside world. At the time, Mombasa was the most advanced township in the region and an important commercial city, with a comparatively better infrastructure, living conditions and healthcare services which may have contributed to lower reported impact. In addition, the higher proportion of non-indigenous population possibly may have meant more past immunity from exposure, such as infection from the first wave, outside of Africa. In contrast, Vanga was less developed and less connected to marine traffic and to the other quickly developing provinces’ districts, which may have contributed to the lower number of cases and deaths reported. Similarly, certain townships in Nyika district reported delayed impact of the pandemic which was attributed to low traffic of marine vessels bringing in infected persons. Clinical reports suggest that the pandemic disproportionately killed the elderly and children, as opposed to the young healthy adults documented elsewhere in the world [32,33], though unfortunately we do not have any data to confirm this. The reported shortfall of labor force during this period may indicate that indeed younger adults were more affected than reported, but a further consideration is that mortality patterns may have varied geographically depending on previous exposure to influenza viruses. One study suggests that in remote populations with little external contact, the mortality among the elderly may have been high [34].

Globally, the pandemic consisted of three successive waves: a mild first wave, the severe second wave and generally lesser severe third wave. All three waves occurred in continental Europe and the Americas [6], whereas the last two in most of Asia and Africa [16]. Countries with all three waves may have suffered a lesser burden than those with only two waves, and it is speculated that the first wave may have induced some immunity [32]. Kenya’s Coastal Province does not seem to have experienced a first wave and therefore may have been more severely affected. Additionally, the precarious living conditions and limited access to care, combined with under-nutrition and armed conflicts, likely increased individual susceptibility to influenza-associated complications in Kenya. Our reported annual mortality rate of 25 per 1000 people is comparable to findings reported elsewhere in continental Africa and in one global estimate [12,17], though other rates in continental Europe, USA, India were lower between 4.8 to 6.5 deaths/1000 population, and China at 8.4–20.1 deaths/1000 population [12]. Of note is that pandemic mortality may have been over-reported because of inclusion of non-pandemic deaths in an era when case definitions were not used and laboratory diagnosis was not available, but the apparent spike in mortality and subsequent drop in 1919 (though not to pre-1918 baseline) does suggest that mortality in Kenya was substantially above other global estimates. 

The 1918 pandemic Influenza had a profound effect on Kenya, possibly more so than in countries in Europe or the Americas because of limited infrastructure, comorbidities and food security. The unprecedented deaths and sicknesses in all races and nationalities additionally may have caused native Africans to doubt the superiority of immigrant settlers and question foreign religions (Christianity and Islamism). This may have led to re-emergence of traditional faith and may have contributed to the struggle for independence [18,35]. However, those with strong Christian faith envisioned the pandemic as a hand of God punishing humanity for its evil for world wars and unjustified invasions [35,36]. 

Our study had limitations. Data were not reported systematically, were incomplete and accuracy was likely unpredictable, and healthcare visits underestimate total morbidity. Some districts provided no numbers of actual suspect cases, only deaths; other districts such as Mombasa may have seriously underreported cases seeking care compared to deaths, which could lead to the high case fatality proportions. Sharp increases in events are likely partly due to increased reporting, either because data were solicited or because people became aware of the severity of the pandemic. Moreover, no consistent age or more granular geographical breakdown were available, nor the detailed evolution of the pandemic impact over time. The data reported were aggregated and became sparse near the end of the pandemic, in early 1919. In 1918, Kenya had no reference national population census data until 1948 [23], therefore, the rates calculated were on aggregate population approximation amalgamated from diverse sources. Moreover, no case definition used to determine influenza illness or death; determination was solely based on physician’s opinion. No detailed clinical information was given and laboratory testing was not available at the time. Interestingly, the separately reported pandemic mortality rate was very similar to the all-cause mortality rate in 1918 which suggests either that non-pandemic deaths were especially underreported that year, or that some of the deaths reported as pandemic deaths were not due to influenza. Similarly, the rate of pandemic cases is substantially higher than the rate of those who sought care for any illness, but the degree of underreporting of pandemic cases that did not seek health care is unknown. The lack of age distribution data meant that we could not examine some epidemiologic hallmarks for the 1918 pandemic such as increased mortality in young adults. Lastly, the experience of the coastal population cannot be reliably extrapolated to inland populations where comorbidities, general health, and previous exposure (therefore immunity) to influenza may have been different [34]. 

Nonetheless, the anecdotal evidence and the sparse data collected paint a picture of a substantial impact of the 1918 pandemic influenza in what is now coastal Kenya, and a sharp reminder that influenza remains a global cause of severe disease in all populations worldwide. Seasonal influenza has been overlooked as a cause of morbidity and mortality in Africa [22], and only recently more data from the African Network of Influenza Surveillance and Epidemiology has shown that influenza viruses circulate widely, year-round and throughout the continent [36]. Nonetheless, even today the lack of consistent vital statistic data and unavailability of diagnostic tests and clinical care in many African settings makes influenza burden estimation challenging. The impact in Africa of the 2009 pandemic, for instance remains debated [21]. 

In conclusion, as we mark 100 years since the emergence of the “Spanish flu”, Kenya and neighboring countries should continue to make efforts to detect, prevent, and respond to influenza to prevent not only morbidity and mortality but also severe social and economic disruption associated with a pandemic. Implementation of a seasonal influenza vaccination program in low-to-middle income countries could mitigate the impact of seasonal influenza. Preparation for the next pandemic requires continued improvement in surveillance, education about influenza vaccines, and efforts to prevent, detect and respond to novel influenza outbreaks.

## Figures and Tables

**Figure 1 tropicalmed-04-00091-f001:**
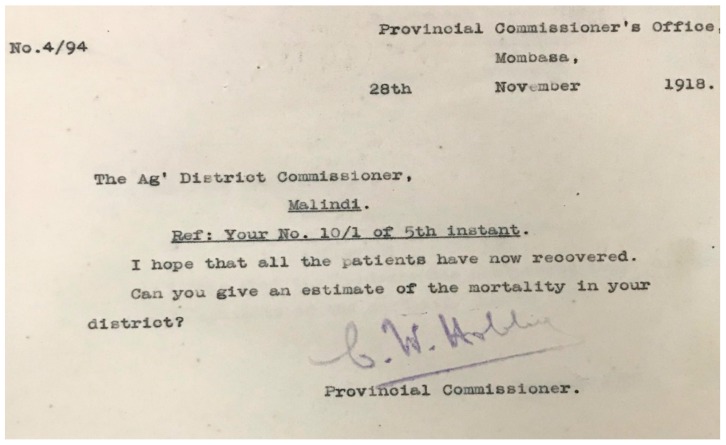
Coast Province administration first letter sent to one of the District Officers requesting the 1918 pandemic influenza deaths. (Source: National Archives Library-KNA, Nairobi Kenya).

**Figure 2 tropicalmed-04-00091-f002:**
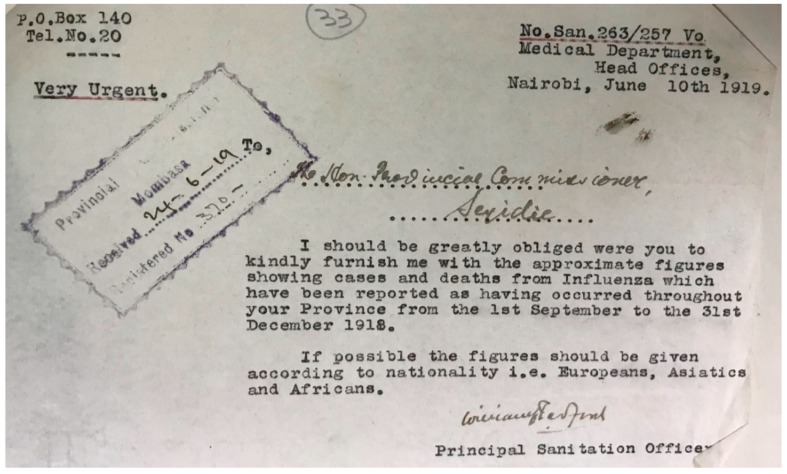
Coast Province administration last letter sent to one of the District Officers requesting the 1918 pandemic influenza cases and deaths. (Source: National Archives Library-KNA, Nairobi Kenya).

**Figure 3 tropicalmed-04-00091-f003:**
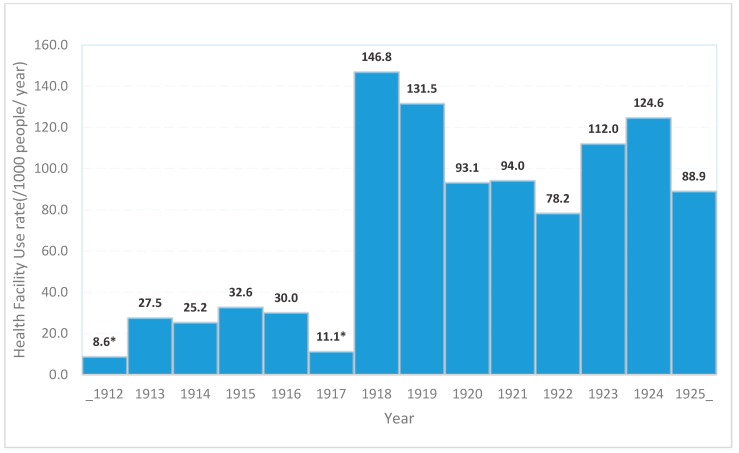
Health facility use rates of Coast Province Kenya, 1912–1925. * Missing data, only Mombasa data reported. Lamu and Tana River districts data missing for years 1917, 1918, 1919 and 1921.

**Figure 4 tropicalmed-04-00091-f004:**
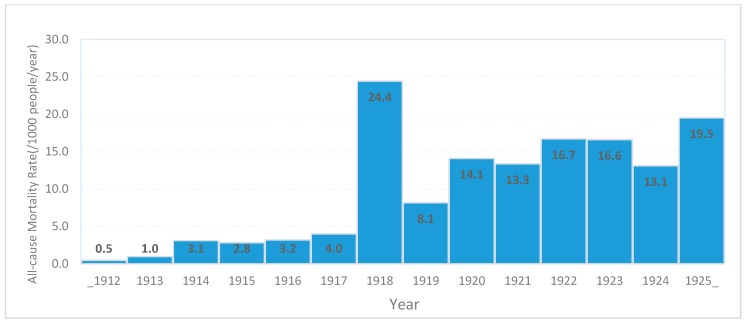
Crude all-cause mortality rates of Coast Province Kenya, 1912–1925. Years 1917, 1918, 1919 and 1921 data for Lamu and Tana River districts missing.

**Figure 5 tropicalmed-04-00091-f005:**
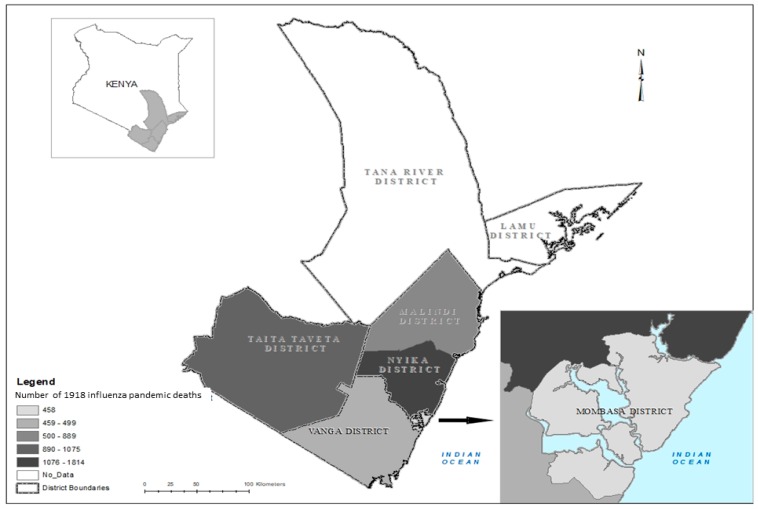
Number of reported 1918 influenza pandemic deaths in districts of Coast Province of Kenya, September 1918 to March 1919.

**Table 1 tropicalmed-04-00091-t001:** Coast Province Kenya population distribution by district and race, 1918 to 1919.

Coast Province Kenya District Name	Population Distribution by District and Race			
	Total N (% total)	Europeans N (%)	Asians/Arabs N (%)	Natives N (%)
1. Mombasa Island	36,824 (20.4)	371 (1.0)	10,774 (29.2)	25,679 (69.6)
2. Vanga	19,761 (10.9)	5 (<1)	423 (2.1)	19,333 (97.0)
3. Taita Taveta	30,545 (16.9)	16 (<1)	135 (<1)	30,394 (99.5)
4. Nyika	71,137 (39.3)	21 (<1)	215 (<1)	70,901 (99.7)
5. Malindi	22,872 (12.6)	13 (<1)	1529 (6.7)	21,330 (93.3)
Total N (%)	181,139 (100.0)	426 (0.3)	13076 (7.2)	167,637 (92.5)

**Table 2 tropicalmed-04-00091-t002:** Morbidity and mortality burden associated with 1918 influenza pandemic in Coastal Kenya Region, September 1918–March 1919.

District Name	Population Estimates	Reported Influenza Cases		Reported Influenza Mortality		Influenza Case Fatality Proportion (%)
		Number	Rate (per 1000)	Number	Rate (per 1000)	
1. Mombasa Island	36,884	1217	33.0	628	17.0	51.6
2. Vanga	19,761	-	-	400	20.2	-
3. Taita Taveta	30,545	10,691	350.0	1065	34.9	9.9
4. Nyika	71,137	20,000	281.1	1700	23.9	8.5
5. Malindi	22,872	-	-	800	35.0	-
Total	181,199	31,908	176.1 *	4593	25.3	10.6 *

* based on data from Nyika, Taita Taveta and Mombasa districts.

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
