# Peer review of "Impact of the 1918 Influenza Pandemic in Coastal Kenya"

_tropicalmed, 2019, doi:10.3390/tropicalmed4020091_

Round 1
Reviewer 1 Report
The authors are to be commended for their careful historical work.
There is a repetition of a quotation in lines 294-297
Author Response
Thank you for the comments. The duplication has been corrected.
Reviewer 2 Report
This is a very well written and interesting paper describing what is clearly a large amount of work to examine the impact of the 1918 influenza pandemic on a province in Kenya. I think this somewhat "atypical" research will be of interest to readers as it not only provides the basic epidemiology of the pandemic in this region, but also provides historic references, quotes, and information from a broader perspective. I enjoyed reading it on multiple levels.
The main concern I have is that I am not clear on how the population denominators were derived. A number of sources were used - did you therefore derive the denominator estimates? Did you take an average from multiple sources for any of the districts? Since you report quite high rates of mortality additional detail and clarity is warranted. If multiple sources reported different population numbers for a district for example, it seems appropriate to report a range of estimates and corresponding rates, i.e., a sensitivity analysis. Depending on the answers to these questions, the Discussion section might need to reflect this issue (e.g., perhaps the population estimates are underestimates).
I also recommend adding in a timeline of the various major events you report, in relation to what was happening in other countries. While details from other parts of the world are provided and you describe numerous specific events such as letters being sent, a figure showing the major "events" on a timeline would be very helpful to follow, for example, the events on page 3 starting at line 113.
Author Response
The main concern I have is that I am not clear on how the population denominators were derived. A number of sources were used - did you therefore derive the denominator estimates? Did you take an average from multiple sources for any of the districts? Since you report quite high rates of mortality additional detail and clarity is warranted. If multiple sources reported different population numbers for a district for example, it seems appropriate to report a range of estimates and corresponding rates, i.e., a sensitivity analysis. Depending on the answers to these questions, the Discussion section might need to reflect this issue (e.g., perhaps the population estimates are underestimates).
Thank you for this comment. We do in fact mention the uncertainty in the section on limitations, but realise that we need more clarity on how the population denominators were calculated. We took a hierarchical approach to population denominators, first looking to Province data then district data (mainly annual reports) and then other sources as available. While this means our rates may be inaccurate, we believe that this would not change over time, so does not overly affect the conclusion of a large spike in morbidity and mortality in 1918. In fact the sources of population denominator from each district when there were multiple were not very different, hence a sensitivity analysis would not provide a great deal of additional meaning with these inherently inaccurate data.
I also recommend adding in a timeline of the various major events you report, in relation to what was happening in other countries. While details from other parts of the world are provided and you describe numerous specific events such as letters being sent, a figure showing the major "events" on a timeline would be very helpful to follow, for example, the events on page 3 starting at line 113.
Thank you for the suggestion – we have included as suggested.
Reviewer 3 Report
This paper is a well-written and welcomed contribution to the scarce literature on the impact and consequences of the 1918 flu in Africa. However, there are some issues that have to be dealt with before this paper can be published.
The origin of the 1918 flu is contested. China (see Nature Medicine, 1999, 5: 384-385) and Europe are other alternatives (see Nature Medicine, 1999, 5: 1351-1352 and 2002, 2, 111-114)
I think your paper shows that it is indeed possible to analyse the impact of the 1918 flu in Africa, if you actually look up the archival material that exists, as you do. I think the lack of data argument made in lines 39-40 is dated and wrong.
On page 2, lines 59-60 you suggest that one reason why your estimate at 2.5% is lower than prior estimates (at 6%; was this an estimate observed or excess mortality?) is that yo subtract the baseline mortality. However, you do not seem to calculate annual mortality in 1918 and 1919 subtracting the baseline mortality. You seem to have data to calculate the excess mortality over time and space - so why not calculate excess mortality?
Line 169, why not also report morbidity rates per 1000 per district here and present those figures in Table 2? Here you should also report on to what degree these figures are close to "real" morbidity. I guess these figures must understate the real morbidity burden as they were based on people seeing a doctor only?
Line 183 - although you do not seem to have data on age-specific mortality - you should refer studies that do have such data (e.g. Geography May Explain Adult Mortality from the 1918-20 Influenza Pandemic, Epidemics 2012 3(1): 46-60) and also use the indirect evidence you have to say something about age-patterns. For example, in line 282, page 7, you report that the pandemic lead to "unprecedented labor shortage". Is not this an indirect evidence that the 1918 flu also in Kenya must have put the largest toll on young adults - the prime of the working age population - and not children and the elderly?
I think you can use the data on ethnic background a little more to substantiate your findings. Why not calculate the row and column percentages in table 1 to show clearer the distributions of the various populations in the 5 districts considered? It will clearly show that mortality was lowest in Mombasa - the district with the largest concentration of non-natives. The non-natives here might have brought with them better protection from prior exposure to similar virus that caused the 1918 flu in places of origin in Europe and Asia (see Epidemics 2012). The presence of Europeans/Asians in this city might also imply that a spring wave was present here - protecting all in this city against subsequent waves. Maybe this major city with heavy contact via trade etc. also had a population with a better average protection from prior exposure to H1N1 virus in the past than the other areas considered?
Line 343: global mortality was 2.5-5.0% using data from Johnson and Mueller, not 0.2%
Figure 3: The heading says "health facility use rates" while in the figure itself it is said "morbidity rates" - please check and correct this - are we talking all diseases/reasons for hospitalizations? Why not calculate excess here as well, as you could do for mortality in figure 4, to better tease out the impact of the 1918 pandemic? And why is the hospitalization rates so high in all years after 1918? What happened after the 1918 flu that kept these figures so high compared to the baseline 1912-1917 of ca. 25 per 1000 per year? Or are the baseline-estimates not correct and too low?
Figure 4. Here, it must be specified that you talk about all-cause mortality (heading within the figure is different from the figure heading). Again, I think you should calculate excess mortality, and I wonder also what kept the figures so high 1918-1925 compared to the baseline of 3 all-cause deaths per 1000 per year? (which is a strangely low figure, by the way. Are they correct?). I also think it is somewhat striking that the estimate for all-cause mortality in 1918 (24.4 per 1000) is so strikingly similar to your influenza mortality per 1000 per the five districts 25.0 per 1000) - Did all die from flu in 1918? Are estimates in figures 4 and table 2 based on the same districts? Lamu and Tana river districts, that you say you lack data from in note to figure 4 is not presented in table 2.
Are morbidity figures in table 3 correct? Only 3.2% sick in Mombasa, and 35% in Taveta and 28% in Nyika - the latter figures seems low, not to talk about the figure for Mombasa. The case fatality ratio is highest for Mombasa, but this district has the lowest mortality of the three. I think these figures for morbidity and case fatality ratios must be treated with extreme caution.
Author Response
The origin of the 1918 flu is contested. China (see Nature Medicine, 1999, 5: 384-385) and Europe are other alternatives (see Nature Medicine, 1999, 5: 1351-1352 and 2002, 2, 111-114)
Thank you for this point. We had used the language first reported in the US, but have further clarified that the actual location of the emergence of the pandemic remains unclear and added a reference for which we thank the reviewer.
I think your paper shows that it is indeed possible to analyse the impact of the 1918 flu in Africa, if you actually look up the archival material that exists, as you do. I think the lack of data argument made in lines 39-40 is dated and wrong.
We agree and have edited the introduction to make this clearer. We are not aware of further archival data that would allow this analysis in other countries in Africa, but are hopeful this would paper would act as an incentive as implied by the reviewer.
On page 2, lines 59-60 you suggest that one reason why your estimate at 2.5% is lower than prior estimates (at 6%; was this an estimate observed or excess mortality?) is that you subtract the baseline mortality. However, you do not seem to calculate annual mortality in 1918 and 1919 subtracting the baseline mortality. You seem to have data to calculate the excess mortality over time and space - so why not calculate excess mortality?
We did not mean to imply that we used excess mortality when comparing with previous work. Our point was that with longitudinal mortality data we can have extra insights. However, we think for the reasons of a very unstable baseline, likely artificially low pre-1918 and increased reporting in unknown ways after 1918, that excess mortality estimates would not bring many advantages. We were fortunate in that there was a specific pandemic death reporting system and we feel that that that is the most accurate estimate. As the reviewer points out, the all-cause mortality data are similar (~25/1000) to the pandemic mortality. This suggests strongly that the baseline all-cause mortality in 1918 was underreported. We have added explanation to the methods to explain this.
Line 169, why not also report morbidity rates per 1000 per district here and present those figures in Table 2? Here you should also report on to what degree these figures are close to "real" morbidity. I guess these figures must understate the real morbidity burden as they were based on people seeing a doctor only?
Yes to last point that these numbers refer to healthcare use or seeing a health care staff in a facility. We do not know how this relates to “real” morbidity in those who did not seek care which are not reported at all. It is correct that those seeking health care are only an unknown fraction of all those who were ill. We have added a phrase in discussion to clarify that healthcare seeking does not equate to overall flu morbidity.
Line 183 - although you do not seem to have data on age-specific mortality - you should refer studies that do have such data (e.g. Geography May Explain Adult Mortality from the 1918-20 Influenza Pandemic, Epidemics 2012 3(1): 46-60) and also use the indirect evidence you have to say something about age-patterns. For example, in line 282, page 7, you report that the pandemic lead to "unprecedented labor shortage". Is not this an indirect evidence that the 1918 flu also in Kenya must have put the largest toll on young adults - the prime of the working age population - and not children and the elderly?
Thank you – we have added to the discussion the possibility of younger adults being more affected than reported an also the point that mortality patterns may be different in different areas, and added the reference for which we thank the reviewer.
I think you can use the data on ethnic background a little more to substantiate your findings. Why not calculate the row and column percentages in table 1 to show clearer the distributions of the various populations in the 5 districts considered? It will clearly show that mortality was lowest in Mombasa - the district with the largest concentration of non-natives. The non-natives here might have brought with them better protection from prior exposure to similar virus that caused the 1918 flu in places of origin in Europe and Asia (see Epidemics 2012). The presence of Europeans/Asians in this city might also imply that a spring wave was present here - protecting all in this city against subsequent waves. Maybe this major city with heavy contact via trade etc. also had a population with a better average protection from prior exposure to H1N1 virus in the past than the other areas considered?
We have added the % as suggested to the table. We agree these are all interesting possibilities and have added some text in the discussion to address, with the caveat that we are limited in the interpretation of these data.
Line 343: global mortality was 2.5-5.0% using data from Johnson and Mueller, not 0.2%
Thank you for catching this error - we have corrected.
Figure 3: The heading says "health facility use rates" while in the figure itself it is said "morbidity rates" - please check and correct this - are we talking all diseases/reasons for hospitalizations? Why not calculate excess here as well, as you could do for mortality in figure 4, to better tease out the impact of the 1918 pandemic? And why is the hospitalization rates so high in all years after 1918? What happened after the 1918 flu that kept these figures so high compared to the baseline 1912-1917 of ca. 25 per 1000 per year? Or are the baseline-estimates not correct and too low?
We have corrected the figure (the within figure title was vestigial and not actually visible in our copy).
We don’t know why mortality remained so high after 1918 and we suspect the pre 1918 baseline is too low. We speculate in the discussion that the difference is most likely due to reporting increases. Mortality rates after 1918 are more consistent (though higher) than current estimates of crude mortality, but it is very difficult to know without a better age structure in 1918 (likely much younger than now) and the cause of death in a pre antibiotic era. We have added a phrase in the discussion to better clarify this point.
Similarly the above is the reason for a reluctance on our part to calculate excess mortality and morbidity since we are not sure what the baseline are and the data are quite unreliable, though useful to illustrate trends and gross effects. We prefer to rely on the direct reports of pandemic mortality than calculate an excess all cause mortality and morbidity.
Figure 4. Here, it must be specified that you talk about all-cause mortality (heading within the figure is different from the figure heading). Again, I think you should calculate excess mortality, and I wonder also what kept the figures so high 1918-1925 compared to the baseline of 3 all-cause deaths per 1000 per year? (which is a strangely low figure, by the way. Are they correct?). I also think it is somewhat striking that the estimate for all-cause mortality in 1918 (24.4 per 1000) is so strikingly similar to your influenza mortality per 1000 per the five districts 25.0 per 1000) - Did all die from flu in 1918? Are estimates in figures 4 and table 2 based on the same districts? Lamu and Tana river districts, that you say you lack data from in note to figure 4 is not presented in table 2.
Thank you. The internal title was removed as was overlooked vestige. We suspect that many of these are data are incorrect and have no easy recourse to verify. Unfortunately we cannot stratify facility use or mortality by district, because most the data in the years of greatest interest are aggregate from provincial reports.
The estimates in figure 4 have data from 2 districts (Lamu and Tana River) missing in the years 1917,1918, 1919 and 1921. This is stated in footnote. Table 2 includes data from 1918-1919 and include the remaining 5 districts. Pandemic data was not available for table 2 from Lamu and Tana River as explained in the methods.
Are morbidity figures in table 3 correct? Only 3.2% sick in Mombasa, and 35% in Taveta and 28% in Nyika - the latter figures seems low, not to talk about the figure for Mombasa. The case fatality ratio is highest for Mombasa, but this district has the lowest mortality of the three. I think these figures for morbidity and case fatality ratios must be treated with extreme caution.
We agree that the nature of these data are very variable, hence highlighting the limitations of long- archived data and have addressed this in the discussion.
Round 2
Reviewer 3 Report
Dear authors!
I liked the revisions you made. I have some minor questions and suggestions for revisions and edits, please see the bubble comments in the attached pdf.
Best,
S-E

Author Response
Thank you for the careful read and review. We have addressed these points. We did not calculate mid year population estimates, since was unclear at times how and when the data on population was collected.
We have attached a further edited version in tracked changes a couple of answers in comments.
